# Nosocomial COVID-19 Infection in a Long-Term Hospital in Spain: Retrospective Observational Study

**DOI:** 10.3390/medicina58050566

**Published:** 2022-04-21

**Authors:** Elena Caro-Martínez, Susana Abad-Collado, Blanca Escrivá-Cerrudo, Shaila García-Almarza, María del Mar García-Ródenas, Elena Gómez-Merino, María-Isabel Serrano-Mateo, Jose-Manuel Ramos-Rincón

**Affiliations:** 1Alicante Institute of Health and Biomedical Research (ISABIAL), 03010 Alicante, Spain; jose.ramosr@umh.es; 2Internal Medicine Department, Sant Vicent del Raspeig Hospital, San Vicente del Raspeig, 03690 Alicante, Spain; sabadcollado@hotmail.com (S.A.-C.); escrivabla@hotmail.com (B.E.-C.); egmerino1@gmail.com (E.G.-M.); serrano_isamat@gva.es (M.-I.S.-M.); 3Geriatric Unit, Sant Vicent del Raspeig Hospital, San Vicente del Raspeig, 03690 Alicante, Spain; shayla.garcia@gmail.com; 4Pneumology Department, Sant Vicent del Raspeig Hospital, San Vicente del Raspeig, 03690 Alicante, Spain; margarciarodenas@gmail.com; 5Internal Medicine Department, Alicante General University Hospital, 03010 Alicante, Spain; 6Clinical Medicine Department, Miguel Hernández University, 03550 Elche, Spain

**Keywords:** nosocomial COVID-19 infection, long-term hospital, mortality, Spain

## Abstract

*Background and Objectives*. The aim of this study is to compare clinical and epidemiological characteristics and outcomes in patients with versus without nosocomial COVID-19 after exposure to SARS-CoV-2 and to analyze the risk factors for severe outcomes of COVID-19 in a long-term hospital in Spain. *Materials and methods*. This retrospective, single-center observational study included all inpatients in a long-term hospital during a COVID-19 outbreak from 21 January to 15 March 2021. *Results*. Of 108 admitted patients, 65 (60.2%) were diagnosed with nosocomial COVID-19 disease (*n* = 34 women (52.3%), median age 77 years). In the univariable analysis, risk factors associated with nosocomial COVID-19 were dementia (OR 4.98 95% CI 1.58–15.75), dyspnea (OR 5.34 95% CI 1.69–16.82), asthenia (OR 5.10, 95% CI 1.40–18.60) and NECesidades PALiativas (NECPAL) (OR 1.28 95% CI 1.10–1.48). In the multivariable analysis, risk factors independently associated with nosocomial COVID-19 infection were dyspnea (aOR 7.39; 95% CI 1.27–43.11) and NECPAL (aOR 1.25; 95% CI 1.03–1.52). Of the 65 patients diagnosed with nosocomial COVID-19, 29 (44.6%) died, compared to 7/43 (16.2%) non-infected patients (OR 4.14, 95% CI 1.61–10.67). Factors associated with mortality in nosocomial COVID-19 were confusion (aOR 3.83; 95% CI 1.03–14.27) and dyspnea (aOR 7.47; 95% CI 1.87–29.82). The NECPAL tool played an important predictive role in both nosocomial COVID-19 infection and mortality (aOR 1.19, 95% CI: 1.00–1.41). *Conclusions*. In a long-term hospital, nosocomial COVID-19 main clinical characteristics associated with infection were dyspnea and NECPAL. Mortality was higher in the group with nosocomial COVID-19; risk factors were confusion and dyspnea. The NECPAL tool may help to predict progression and death in COVID-19.

## 1. Introduction

In December 2019, a series of viral pneumonia cases was reported in China. The World Health Organization (WHO) named the novel coronavirus “severe acute respiratory syndrome coronavirus 2” (SARS-CoV-2) as the causative virus of coronavirus disease 2019 (COVID-19) [1]. On January 2020, WHO declared that this viral outbreak constituted a public health emergency of international concern, foreshadowing the COVID-19 pandemic and the high number of confirmed infections and deaths [2]. Cases have been confirmed in over 180 countries and regions around the world [3]. The disease has a high case-fatality rate and severity, particularly among older people [1], but all age groups are susceptible [4].

In Spain, long-term hospitals are designed to care for mainly older, chronic, pluripathological patients. The clinical status of these patients may lead to an increased risk of serious SARS-CoV-2-related complications [5]. Nosocomial COVID-19 infection is defined as an infection acquired in hospital by a patient who was admitted for a different reason (at least 15 days prior to a positive COVID-19 diagnosis).

Here, we report a nosocomial COVID-19 outbreak in a long-term hospital. The aim was to compare clinical and epidemiological characteristics and outcomes in patients with versus without nosocomial COVID-19 after exposure to SARS-CoV-2. The secondary aim was to assess risk factors for severe outcomes of COVID-19.

## 2. Materials and Methods

### 2.1. Design and Participants

This retrospective, single-center observational study included all hospitalized patients in the Sant Vicent del Raspeig hospital in Alicante (Spain) during a COVID-19 outbreak from 21 January to 15 March 2021. The setting was a long-term hospital for chronically ill patients with four wards: long-stay, convalescence, brain injury, and palliative care.

When the first positive case of SARS-CoV-2 was detected, a mass screening campaign was implemented based on real-time reverse transcription (RT-PCR) for SARS-CoV-2 on days 1, 5, and 9 in all patients and active health personnel who had not already tested positive. RNA was extracted from oropharyngeal swabs. It was considered nosocomial COVID-19 infection if the admission of the patient to the hospital was more than 48 h and if the patient’s oropharyngeal swab on day 1, 5, or 9 was positive and the patient developed any COVID-19 symptom. The patients who were positive were transferred to the COVID-19-enabled ward following isolation and treatment protocols. No further patients were admitted to the hospital until the outbreak was brought under control on 28 February 2022.

At the time the first COVID-19 patient tested positive, 19 January 2021, there were 141 patients admitted to the hospital. Initially, 3 of them were transferred to an acute hospital for treatment. Twenty-two patients were discharged early or died before the start of the screening. Eight patients were negatively discharged after screening and were admitted a few days later to another hospital with a diagnosis of COVID-19. Finally, 108 patients that were screened could be followed up until discharge from the hospital (Figure 1).

### 2.2. Data Collection

The demographic and clinical characteristics, including signs and symptoms, radiographic parameters, comorbidities and complications, clinical treatment, and outcomes were extracted from electronic health records. The laboratory values included C-reactive protein (CRP; reference values < 5 mg/L); ferritin; lactate dehydrogenase; glomerular filtration rate; and lymphocyte count. 

We also used the following scales: the Clinical Frailty Scale (CSF) to measure frailty (range 1–9, with lower scores indicating less frailty) [5]; the Charlson Comorbidity Index (CCI) to predict 10-year survival in patients with multiple comorbidities (19 items, which if present, influence life expectancy) [6]; the Barthel Index for activities of daily living, to assess functional independence (range 0–100, the lower scores indicating higher dependency) [7]; the Pfeiffer test to assess organic brain deficit in elderly patients (range 0–10 errors, the more errors indicating more severe cognitive impairment) [8]; the Emina scale to evaluate the risk of developing pressure ulcers (range 0–15, the lower scores indicating lower risk) [9]; the Confusion Assessment Method (CAM) to identify and recognize delirium quickly and accurately according to four diagnostic features (acute onset and fluctuating course, inattention, disorganized thinking, and altered level of consciousness) [10]; the NECesidades PALiativas (NECPAL CCOMS-ICO) scale for the early identification of people with palliative care needs and a life-limiting prognosis, combining the physician’s own insight with objective indicators of disease progression and indicators of chronic advanced conditions [11]; the Controlling Nutritional Status (CONUT) score to identify undernourished patients in the hospitalized population, with the score derived from the values of serum albumin, total cholesterol, and lymphocyte counts [12]; the Morse fall scale to assess a patient’s likelihood of falling (score 0–125, with >45 indicating a high risk of falling) [13]; and a visual analog scale (VAS) for rating pain on a scale of 0 to 10 (higher scores indicating more pain) [14].

### 2.3. Statistical Analysis

For clinical data, categorical variables were expressed as absolute number and proportion. The continuous variables were expressed as medians and interquartile range (IQR), and the discrete variables were expressed as mean and standard deviation (SD). The categorical variables were analyzed using the chi-squared test or Fisher’s exact test, and continuous variables were analyzed using the Mann–Whitney U test. The risk factors associated with mortality due to COVID-19 were estimated using multivariable logistic regression, and the effect sizes were expressed as adjusted odds ratios (aOR) with a 95% confidence interval (CI). 

In the analysis of the factors related to nosocomial COVID-19 and mortality, the statistically significant variables identified in univariable analyses (*p* < 0.05) were entered into a multivariable logistic regression using a stepwise forward selection method with the likelihood ratio test. Following the thump rule, a ratio of at least 10:1 was used for the sample size and the number of variables. Model validity was evaluated using the Hosmer–Lemeshow test for estimating the goodness-of-fit to the data, and its discriminatory ability used the area under the curve (AUC). Variables that were missing for more than 25% of patients were excluded from the analysis. The significance level was established at *p* < 0.05. Statistical analyses were conducted using SPSS software, version 23.0 (IBM Corp., Armonk, NY, USA).

### 2.4. Ethical Aspects

The study was conducted in accordance with the Declaration of Helsinki. The Ethical Committee of Alicante General University Hospital approved the research protocol (Ethics Committee code: CEIm PI2021-100, date of approval: 1 July 2021) and waived the need for informed consent because it did not constitute a clinical study, according to national and European regulations.

## 3. Results

The initial case, or index patient of nosocomial COVID-19, was reported on 18 January 2021. In total, 108 patients were screened and followed up until discharge from the hospital; of these, 65 (60.2%) were diagnosed with nosocomial COVID-19 disease, and 43 patients remained negative (Figure 1). Moreover, a total of 330 orderlies, nurses, nursing assistants, students, doctors, and administrative staff were tested at the time of the outbreak. A total of 70 were positive (21.2%). The most affected were nursing assistants (34.7%), followed by orderlies (13.9%), nursing students (11.1%) and administration workers (1.4%). For the doctors, 3 out of 25 (12.5%) were infected. 

In total, there were 65 patients who acquired nosocomial COVID-19 (median age 77 years, 34 (52.3%) women). Five patients (7.7%) had been institutionalized prior to admission. All transferred patients had negative SARS-CoV2-PCR-RT in the 48 h before transferring to our hospital. Among the associated comorbidities, the most frequent were arterial hypertension (76.9%), cerebrovascular disease (36.9%), and diabetes mellitus (35.4%) (Table 1). The median CCI was 6.1 (IQR 4–9).

The differences in the profile of the patients infected versus not infected with nosocomial COVID-19 disease appear in Table 1. CCI was higher in the COVID-19 group (*p* < 0.001), but dementia was the only specific comorbidity associated with an infection (33.8% vs. 9.3%, *p* < 0.001).

At the time of SARS-CoV2 screening, no patients were symptomatic, but those with positive PCRs went on to develop symptoms, principally hypoxemia (97% vs. 3%, *p* < 0.001), dyspnea (35.4% vs. 9.3%, *p* < 0.001), and asthenia (27.7% vs. 7%, *p* < 0.001). Radiological findings on a chest X-ray were normal in 45.5% of patients in the COVID-19 group and 56% in the non-infected patients; 14 (25.5%) patients with COVID-19 developed pneumonia.

The COVID-19 group also received more antibiotic treatment (49.2% vs. 16.3%), corticosteroids (58.5% vs. 11.6%), and ventilatory support (69.2% vs. 33.3%). Patients infected with nosocomial COVID-19 presented significantly higher NECPAL (*p* < 0.001) and CAM scores (*p* = 0.02), along with a lower Barthel index (*p* = 0.011) (Table 2).

In univariable analyses, risk factors associated with nosocomial COVID-19 infection were dementia (OR 4.98 95% CI 1.58–15.75; *p* = 0.003), NECPAL (OR 1.28 95% CI 1.10–1.48; *p* = 0.001), dyspnea (OR 5.34 95% CI 1.69–16.82; *p* = 0.002) and asthenia (OR 5.10, 95% CI 1.40–18.60; *p* = 0.008). Following the thump rule, a ratio of at least 10:1 for the sample size and the number of variables, one variable was chosen for every 10 events, including age and gender. In multivariable analyses, risk factors associated with nosocomial COVID-19 infection were dyspnea (aOR 7.39, 95% CI 1.27–43.11; *p* = 0.026) and NECPAL score (aOR 1.25, 95% CI 1.03–1.52; *p* = 0.032). In this model, the *p* value for the Hosmer–Lemeshow goodness-of-fit test was 0.83 with an AUC of 0.792 (95% CI 0.696–0.887), which indicates good predictive ability.

There was a significant difference in the mortality between groups. Of the 65 patients who were diagnosed with nosocomial COVID-19 infection, 29 (44.6%) died compared to 7/43 (16.2%) non-infected patients (OR 4.14, 95% CI 1.61–10.67; *p* < 0.001). The mortality due to infectious diseases other than COVID-19 was significantly higher in the non-infected group (*p* = 0.027) (Table 3).

Table 4 shows significant clinical characteristics in patients with nosocomial COVID-19, according to whether they survived the study period. In multivariable analyses, the variables associated with mortality were confusion (aOR: 3.84; 95% CI: 1.03–14.26, *p* = 0.045), dyspnea (aOR 7.47, 95% CI: 1.87–29.83, *p* = 0.004) and NECPAL (aOR 1.19, 95% CI: 1.04–1.41, *p* = 0.045). In this model, the *p* value for the Hosmer–Lemeshow goodness-of-fit test was 0.46 with an AUC of 0.811 (95% CI 0.708–0.913), indicating good predictive ability.

## 4. Discussion

This retrospective cohort study reported outcomes in patients admitted to a long-term hospital during a SARS-CoV-2 outbreak in which 60% of the patients were infected. It is important to highlight that, in our study, the people at the highest risk of acquiring nosocomial COVID-19 infection were those with palliative needs and greater comorbidity. This group showed 20% higher mortality than the non-infected group. Mortality among infected patients was related to palliative needs, the presence of confusion at diagnosis, and dyspnea. These parameters should be taken into account when establishing the diagnosis and evolution of nosocomial COVID-19.

The proportion of nosocomial COVID-19 infections in our center was higher than in previous studies [15]. Public Health England (PHE) estimated the rate of nosocomial infections to be 10% to 22%, [16], whereas a meta-analysis of Chinese data estimated the risk at 44% [17].

This study identified several risk factors for acquiring nosocomial COVID-19 in a long-term hospital, namely underlying dementia and a higher Charlson comorbidity index. NECPAL, Barthel, and CAM scores also showed important associations. Previous studies have also found an association between dementia and a higher risk of COVID-19, and these patients are more likely to have severe or fatal disease compared to people without dementia, even after adjusting for age, sex, institutionalization, and preexisting conditions [18,19]. Patients with dementia were almost four times more likely to die from COVID-19 than patients without dementia [20,21]. Most of our patients had low life expectancy, and this was even lower in the COVID-19-infected group. These patients also showed a greater need for palliative care (NECPAL), greater limitations for performing basic activities of daily living (Barthel Index) [7], and a higher risk of developing delirium (CAM) [10]. Indeed, the Barthel Index has proven a good predictor of mortality in geriatric inpatients [21]. Increased mortality in elderly patients has been associated with comorbidities, especially dementia and cerebrovascular disease, as well as frailty, associated with a poorer immune response [20]. Frailty, malnutrition, and other geriatric syndromes are important factors for assessing elderly patients at a high risk of decompensation of their underlying pathology, constituting a high-risk group for COVID-19 infection.

Our results show higher mortality rates in patients that acquired COVID-19 in the hospital than in hospitalized patients who were not infected; however, the latter group showed higher mortality due to other infectious diseases. The factors associated with mortality in nosocomial COVID-19 patients were confusion and dyspnea. Dyspnea has also been associated with mortality in many reports [22,23]. Other important factors independently associated with COVID-19 mortality in the literature included advanced age, increased C-reactive protein, reduced renal function, coronary artery disease, increased frailty [16], and poor preadmission functional status [24,25,26]. Mortality was 1.3 times greater in immunosuppressed patients with nosocomial compared to community-acquired infection [17]. Our results show that the NECPAL tool is associated with mortality in nosocomial COVID-19 patients. The tool was created to identify patients with chronic disease and a limited life expectancy in clinical practice who might benefit from palliative care [11]. The tool has been widely used in clinical practice in different countries and is currently available in several languages [27,28,29]. Previous studies showed that the NECPAL tool can help predict mortality in patients with advanced chronic conditions [27,28,29]. Taken together, this evidence illustrates the intricate relationship by which the nosocomial circulations of SARS-CoV-2 and comorbidities contribute to increasing the risk of mortality.

By implementing the rapid establishment of an expanded SARS-CoV-2 screening program in healthcare workers, we observed that the time from positive PCR for SARS-CoV-2 to initial symptoms was more than seven days and that the predominant symptom was hypoxemia and asthenia rather than dyspnea. In fact, hypoxemia and asthenia were also risk factors for nosocomial COVID-19 in our long-term hospital. These findings are difficult to contextualize due to the paucity of point-prevalence data from asymptomatic individuals in similar healthcare settings or the wider community.

For contrast, 60% of asymptomatic residents in one study tested positive in the midst of a care home outbreak [26]. Temperature did not play a fundamental role in monitoring symptoms in this group. This could be because, in these patients, the prompt recognition of COVID-19-like symptoms probably occurred during the daily inpatient assessment, leading to a quick laboratory and clinical diagnosis of a COVID-19 infection.

The outbreak finished on 15 March 2021, after the last patient with nosocomial COVID-19 was discharged. At the time of this writing, no other coronavirus outbreak has been detected in our hospital. With low hospital-acquired infection rates, this study demonstrates that effective infection control policies are in place. A review [30] identified and systematically synthesized the findings of studies that consistently reported the benefit of contact tracing and screening for preventing COVID-19, which is the best available evidence that policy makers and implementers can use in the process of infection prevention interventions.

The main strength of this study is that it analyzes the experience of a long-term hospital that strives to remain COVID-19-free to protect its frail patient population. This study analyzed many scores (Charlson comorbidity index, CAM, and Barthel) and highlights the importance of NECPAL as a tool to promptly identify people with palliative care needs and life-limiting prognosis, including those with COVID-19. This report can provide insight relevant to future nosocomial outbreaks in other long-stay hospitals.

The limitations of the study include its focus on hospitalized patients during an outbreak in the third wave of the pandemic. In addition, the study is set in a long-term hospital, and it is important to recognize that data from this single center may not be applicable to the general population due to differences in patient profiles, available healthcare resources, and the preparedness of healthcare providers to respond to overwhelming demands on services.

## 5. Conclusions

In conclusion, our study showed that COVID-19 can play an important role in the death of patients hospitalized in a long-term care center. The clinical presentation of nosocomial COVID-19 infection has a later onset, with milder and atypical symptoms in these patients. Dyspnea and NECPAL are risk factors for acquiring a COVID-19 infection, and confusion and dyspnea are risk factors for COVID-19 mortality. The NECPAL tool may also help to evaluate the risk of progression and death from COVID-19. The implementation of screening is essential for the control of a nosocomial outbreak. Further research is needed to investigate the pathogenesis of these outcomes in COVID-19 illness. We recommend that clinicians, ethicists, and policymakers consider these empirical findings.

## Figures and Tables

**Figure 1 medicina-58-00566-f001:**
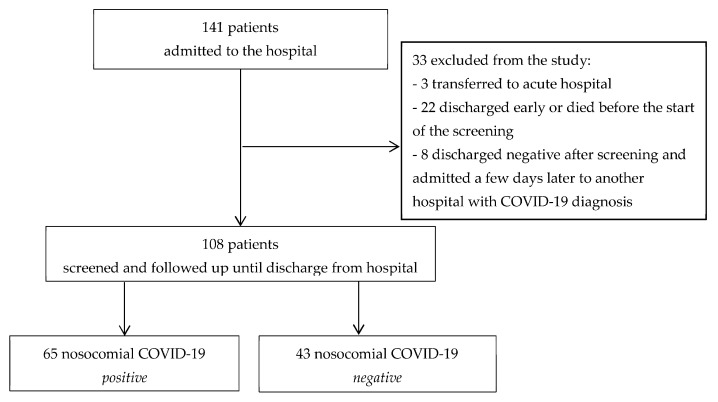
Protocol of the screened patients for their inclusion in the study.

**Table 1 medicina-58-00566-t001:** Epidemiological and clinical differences in patients with and without nosocomial COVID-19.

	COVID-19 (*n* = 65)	No COVID-19 (*n* = 43)	*p* Value
Age, years, median (IQR)	81 (70–89)	73 (65–84)	0.092
Sex, *n* (%)			0.12
Female	34 (52)	27 (63)	
Male	31 (48)	16 (37)	
Hospitalization units, *n* (%)			0.72
Convalescence	37 (57)	28 (65)	
Long-stay	15 (23)	8 (19)	
Palliative care	8 (12)	3 (7)	
Brain injury	5(8)	4 (9)	
Charlson (score), median (IQR)	6.0 (5–9)	5 (3–7)	0.028
Comorbidities, *n* (%)			
Hypertension	50 (76.9)	27 (62.8)	0.11
Cerebrovascular disease	24 (36.9)	10 (23.3)	0.13
Diabetes mellitus	23 (35.4)	15 (34.9)	0.96
Dementia	22 (33.8)	4 (9.3)	0.003
Lung disease	20 (30.8)	9 (20.9)	0.26
Heart disease	17 (26.2)	12 (27.9)	0.84
Kidney disease	15 (23.1)	11 (25.6)	0.77
Symptoms, *n* (%)			
Cough	49 (75.4)	37 (86)	0.18
Hypoxemia *	32 (97)	1 (3)	<0.001
Dyspnea	23 (35.4)	4 (9.3)	0.002
Confusion/disorientation	21 (32.3)	7 (16.3)	0.063
Asthenia	18 (27.7)	3 (7)	0.008
Fever	4 (6.2)	3 (7)	0.87
Constants in PCR screening			
Baseline SatO_2_, median (IQR)	96 (95–98)	97 (96–98)	0.11
Heart rate, bpm, mean (SD)	79 (16.60)	78 (13.01)	0.84
Temperature, °C, mean (SD)	36.53 (0.36)	36.46 (0.51)	0.44
Systolic blood pressure, mmHg, mean (SD)	122 (19.82)	127 (15.84)	0.10
Diastolic blood pressure, mmHg, mean (SD)	68 (11.97)	72 (13.20)	0.17
Analytical results			
Glomerular filtration rate, mL/min, median (IQR)	80.9 (47.54–90)	80.2 (18.17–90)	0.86
C-reactive protein, mg/L, median (IQR)	3.75 (1–10.67)	0.43 (1.61–7.32)	0.33
Lactate dehydrogenase, UI/L, median (IQR)	258 (197–307)	228 (194–285)	0.94
Lymphocytes, ×10^3^, mean (IQR)	1358 (801)	1537 (710)	0.24
Ferritin, µg/L, mean (SD)	997 (1726)	563 (564)	0.14
Chest X-ray, *n* (%)			0.21
Normal	25 (45.5)	14 (56)	
Unilateral pneumonia	14 (25.5)	4 (16)	
Bilateral pneumonia	9 (16.4)	1 (4)	
Treatment, *n* (%)			
Antibiotics	32 (49.2)	7 (16.3)	0.001
Corticosteroids	38 (58.5)	5 (11.6)	<0.001
Ventilatory support	45 (69.2)	14 (33.3)	<0.001
Oxygen therapy	42 (93.3)	14 (93.3)	0.71
Mortality, *n* (%)	29 (44.6)	7 (16.3)	0.002

* <94% or decrease in SatO_2_; IQR: interquartile range; SD: standard deviation; bpm: beats per minute.

**Table 2 medicina-58-00566-t002:** Differences in comorbidity, frailty, and dependence scales in patients with versus without nosocomial COVID-19.

	COVID-19 (*n* = 65)	No COVID-19 (*n*= 43)	*p* Value
NECPAL, median (IQR)	6 (2–8.5)	2 (0–4)	<0.001
Barthel, median (IQR)	10 (0–20)	15 (5–52.5)	0.011
CAM, median (IQR)	3.5 (0–7)	0 (0–2)	0.020
Clinical Frailty Scale, median (IQR)	7 (6.5–8)	7 (5–8)	0.064
EMINA scale median (IQR)	8 (7–10)	7 (5–8)	0.11
CONUT, median (IQR)	5 (4–7)	5 (2–7)	0.31
VAS, median (IQR)	0 (0–1)	1 (0–1)	0.46
Pfeiffer, median (IQR)	5 (1–9)	6 (2–9)	0.67
Morse fall scale, median (IQR)	50 (35–65)	55 (35–75)	0.88

IQR: interquartile range; CI: confidence interval. NECPAL: NECesidades PALiativas; Barthel: 0–20 total dependency, 21–60 severe, 61–90 moderate, 91–99 slight dependency; CAM: confusion assessment method (fluctuating course, inattention, disorganized thinking, and altered level of consciousness); Clinical Frailty Scale: 1–3 robust, 4 pre-frail, 5 mildly frail, 6 moderately frail, 7–8 severely frail, 9 terminally ill. EMINA, risk of pressure ulcers: 0 no risk, 1–3 low risk, 4–7 moderate risk, 8–45 high risk; CONUT, Nutritional Control Index: 0–1 normal, 2–4 light, 5–8 moderate, 9–12 severe; VAS: visual analogue scale 0–10 no pain-the worst pain. Pfeiffer Test: 0–10 errors; more errors = more severe cognitive impairment. Morse fall scale: 0–24 low, 25–44 moderate, >45 high risk of falling.

**Table 3 medicina-58-00566-t003:** Cause of death in patients with 65 nosocomial COVID-19 and 43 without nosocomial COVID-19.

	COVID-19 *n* = 29 (44.6%) *n* (%)	No COVID-19 *n* = 7 (16.2%) *n* (%)	*p* Value
COVID-19	9 (31)	0 (0)	0.16
Cardiovascular diseases	5 (17.2)	1 (14.3)	1.00
Neoplasm	5 (17.2)	2 (28.6)	0.60
Infectious disease other than COVID-19	4 (13.8)	4 (57.1)	0.027
Advanced dementia	3 (10.3)	0 (0)	1.00
Severe obstructive lung disease	1 (3.4)	0 (0)	1.00
Chronic end-stage kidney disease	1 (3.4)	0 (0)	1.00
Cirrhosis	1 (3.4)	0 (0)	1.00

**Table 4 medicina-58-00566-t004:** Significant clinical characteristics in patients with nosocomial COVID-19 infection according to survival.

	Died (*n* = 29)	Survived (*n* = 36)	Crude OR (95% CIs)	*p* Value
Age, years	82 (73–90)	80.5 (64–86.5)	0.98 (0.94–1.01)	0.24
*Sex, n (%)*				0.36
Female	17 (59)	17 (49)	1	
Male	12 (41)	19 (53)	0.63 (0.23–1.70)	
*Scales*				
NECPAL, median (IQR)	4.49 (4.4)	3.2 (3.11)	1.20 (1.05–1.39)	0.009
*Symptoms, n (%)*				
Hypoxemia *	19 (65.5)	13 (36.1)	3.36 (1.21–9.36)	0.018
Dyspnea	16 (55.2)	7 (19.4)	5.09 (1.69–15.37)	0.003
Confusion/disorientation	15 (51.7)	6 (16.7)	5.36 (1.71–16.74)	0.003

* <94% or decrease in SatO_2_. OR: odds ratio; CI: confidence interval; IQR: interquartile range; NECPAL: NECesidades PALiativas.

## Data Availability

ECM and JMRR have full access to the data and are the guarantors of the data.

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
