# Peer review of "Nosocomial COVID-19 Infection in a Long-Term Hospital in Spain: Retrospective Observational Study"

_medicina, 2022, doi:10.3390/medicina58050566_

Round 1

Reviewer 1 Report

The authors may consider to reduce the content of the tables. Most of the tables contain too many parameters with very few number of subjects.  That much details are not warranted for this manuscript.  Minor groups could be merged to make them more comprehensible.

Reviewer 2 Report

I have a concern about the data limited in this study. 

Reviewer 3 Report

I read the edits with joy. My opinion is that the manuscript can be published in the present form in Medicina in the current form.

Author Response

This manuscript is a resubmission of an earlier submission. The following is a list of the peer review reports and author responses from that submission.

Round 1

Reviewer 1 Report

The number of patients in the study is insufficient to draw conclusions. The method in the study is unclear, patient selection is not explained in detail, and the description is not satisfactory. This study, which has very little scientific contribution under these conditions, will come to a level that will contribute scientifically with a larger number of patients and corrected method selection. Best regards

Author Response

Dear reviewer,

Please see the two attached files

1. The response to your comments
2. The final article with the changes made

Reviewer 2 Report

  1. The manuscript is clearly written, apart from a few minor spelling errors (eg: page 3, line 83).
  2. Although the term nosocomial appears in the title, how the patients were considered as nosocomial or not is not that clear from the manuscript. 
  3. Rather than establishing this clear distinction and exploring the established risk factors for nosocomial infection, the authors go on to compare the groups with so many scales. 
  4. The study sample doesn't allow for this many comparisons, please remember the thump rule of 10:1 samples for every variable that is being analyzed. It seems like that authors are hunting for p values, it would be nice if the authors bring down the number of co-variables and come up with a more realistic estimates.
  5. Many a known risk factors of nosocomial transmission, like the number of procedures, duration and proximity of exposure, attending clinical staff or a care giver turning positive, etc are missing. 
  6. The authors seems to use ORa for adjusted Odds Ratio in tables.  It is suggested that better they use one of the standard formats (aOR or adjOR), and ensure that the full form is clearly spelled out at the first instance.
  7. The percentage estimates on Table 4 appear to be miscalculated.

Reviewer 3 Report

The aim of this paper was to compare clinical 16 and epidemiological characteristics and outcomes in patients with versus without nosocomial 17 COVID-19 after exposure to SARS-CoV-2 in a long-term hospital in Spain.

The paper needs more proofreading. and the cited work are up to date. However, there are some important points that should be addressed by the authors. Details are given below

  • Along illustration in the abstract about the results for the experiments studies. Try to reduce this and work on that in the result section
  • The caption for figure 1 in another page for the figure .Try to resolve this
  • The data sample size 108 is very limited. Try to increase it
  • There are a lot of missed information in table 3. Why this is missed?
  • Section 6 -- patents? this section is empty? What will you do in this section?
  • Add discussion section and show the results how it is important in the diagnosis for COVID-19

Reviewer 4 Report

Line 186 (Table 3)

Table 3 suggests that dementia is a statistically significant risk factor. I was wondering if this result is due to a possible confounder such as old age. In other words, were patients with dementia significantly older than those who did not have dementia?
